# Systemic Characterization of the Gut Microbiota Profile after Single Mild Ischemic Stroke and Recurrent Stroke in Mice

**DOI:** 10.3390/biomedicines12010195

**Published:** 2024-01-16

**Authors:** Decao Yang, Panxi Sun, Yong Chen, Haojie Jin, Baohui Xu, Qingbian Ma, Lixiang Xue, Yan Wang

**Affiliations:** 1Institute of Medical Innovation and Research, Peking University Third Hospital, Beijing 100191, Chinalixiangxue@hsc.pku.edu.cn (L.X.); 2Medical Research Centre, Peking University Third Hospital, Beijing 100191, China; 3Ministry-of-Education Key Laboratory of Xinjiang Endemic and Ethnic Diseases, School of Medicine, Shihezi University, Shihezi 832099, China; 4Department of Neurology, Peking University Third Hospital, Beijing 100191, China; 5The Key Laboratory for Silviculture and Conservation of Ministry of Education, The College of Forestry, Beijing Forestry University, Beijing 100107, China; 6Department of Surgery, Division of Vascular Surgery, Stanford University School of Medicine, Stanford, CA 94305, USA; baohuixu@stanford.edu; 7Department of Emergency Medicine, Peking University Third Hospital, Beijing 100191, China; 8BioBank, Peking University Third Hospital, Beijing 100191, China

**Keywords:** ischemic stroke, recurrent stroke, gut microbiota, metabolism

## Abstract

It has been estimated that one in four stroke patients may have recurrent stroke within five years after they experienced the first stroke. Furthermore, clinical studies have shown that recurrent stroke negatively affects patient outcomes; the risk of disability and the death rate increase with each recurrent stroke. Therefore, it is urgent to find effective methods to prevent recurrent stroke. The gut microbiota has been proven to play an essential role after ischemic stroke, while sudden ischemia disrupts microbial dysbiosis, and the metabolites secreted by the microbiota also reshape the gut microenvironment. In the present study, we established a recurrent ischemic mouse model. Using this experimental model, we compared the survival rate and ischemic infarction between single MCAO and recurrent MCAO, showing that, when two surgeries were performed, the mouse survival rate dramatically decreased, while the infarction size increased. Fecal samples were collected on day 1, day 3 and day 7 after the first MCAO and day 9 (2 days after the second MCAO) for 16S sequencing, which provided a relatively comprehensive picture of the microbiota changes. By further analyzing the potential metabolic pathways, our data also highlighted several important pathways that were significantly altered after the first and recurrent stroke. In the present study, using an experimental mouse model, we showed that acute ischemic stroke, especially recurrent ischemia, significantly decreased the diversity of the gut microbiota.

## 1. Introduction

Ischemic stroke is one of the leading causes of disability and death worldwide.

Based on the past 20 years of evidence, ischemic stroke has been shown to be not only associated with severe brain damage but also correlated with the risk of stroke recurrence [1,2,3]. It has been shown that, in a Canadian cohort, patients receiving secondary stroke prevention had a 40% decreased risk of death, recurrence or vascular events [4]. Similarly, in a British cohort, the downregulated recurrence rate facilitated a reduction in mortality [5], indicating that well managing the rate of recurrent stroke is a key factor in controlling the overall death rate of ischemic stroke.

The gut has been termed “the secondary brain”, as accumulating evidence shows that gut microbes and their secreted metabolites closely participate in communication with the brain. For instance, the gut microbiota can influence the development and pathology of the gut and central nervous system, such as by regulating neurotransmitters. Preliminary studies have shown that microbiota-based interventions can alter neurotransmitter levels and may have some impact on host physiology [6]. Many recent studies have shown that an altered gut microbiota is involved in many neurological diseases, including autism, anxiety, obesity, schizophrenia, Parkinson’s disease, Alzheimer’s disease and ischemic stroke [7]. It has been shown that ischemic stroke can induce the dysbiosis of the gut microbiota and epithelial barrier integrity [8]. Based on published clinical data, more than 60 microbes were reported to be upregulated (e.g., *Lactobacillus*), and approximately 30 microorganisms (e.g., *Roseburia*) were downregulated in stroke patients. Using an ischemic stroke mouse model, Spychala et al. showed that the gut microbiota was also closely related to age. Transferring the microbiota from aged mice to young mice decreased mouse behavior performance, increased inflammatory cytokines and increased mortality [9]. In addition, different sex-derived gut microbiota could influence stroke outcomes by mitigating inflammatory responses [10]. The gut microbiota of female mice has been shown to contain lower levels of systemic proinflammatory cytokines after ischemic stroke [10].

Microbiota-derived metabolites can directly or indirectly interact and communicate with brain cells. For instance, gut microbes can communicate with the brain through short-chain fatty acids, which are abundant in the gut and connected to the brain via the vagus nerve to have a direct impact on the brain [11]. Trimethylamine N-oxide (TMAO) is another well-known metabolite that leads to cardiovascular events, including stroke, according to 17 clinical studies involving more than 25,000 subjects [12]. In an experimental stroke model, mice receiving more TMAO-producing microbiota from human subjects had worse stroke outcomes, indicating the essential role of gut microbiota metabolites.

Although many investigations have been performed based on clinical studies or experimental animal models to explore the gut–brain axis after ischemic stroke, until now, very few studies have focused on recurrent ischemic stroke. Thus, in the current study, we established an experimental animal model of stroke recurrence. Using this model, fecal samples were collected at various time points to provide a systematic picture of how the microbiota changes after the first and second strokes and the potential influences on metabolic pathways.

## 2. Materials and Methods

### 2.1. Animals

Male C57BL/6J mice at 10 weeks of age were housed at the Department of Laboratory Animal Science, Peking University Medical Center. Five mice were housed in each cage under a 12:12 h light–dark cycle at a temperature of 25–27 °C and 40–60% humidity, with freely available food and water. All animal experiments complied with the ARRIVE guidelines and were carried out in accordance with the National Institutes of Health Guide for the Care and Use of Laboratory Animals.

### 2.2. Induction and Assessment of Ischemic Stroke

A total of 36 male mice were included in this study. The animals were used and randomized to three subgroups: (1) a sham control group (*n* = 10); (2) a group with a single mild stroke (*n* = 10); and (3) a group with a recurrent stroke, having a mild stroke followed by another stroke 6 days later (*n* = 16, including dead animals). The mice first had mild MCAO. The animals were subjected to a single mild MCAO procedure (30 min), and the infarction size was measured using TTC staining. The secondary MCAO surgery was performed on day 7 to mimic recurring MACO (Figure 1A) [1]. All the mice collected for microbiota screening met the following criteria: (1) had no neurological deficits after stroke; (2) had evidence of surgical subarachnoid hemorrhage; and (3) did not fit the baseline of the behavioral test before surgery for that test.

### 2.3. Infarction Measurement

After ischemia, the mice were deeply anesthetized with isoflurane and euthanized at designated time points. Their brains were sliced into five slices of 2 mm thickness and stained in 2% 2,3,5-triphenyltetrazolium chloride (TTC staining, Cat^#^ T8877, Sigma Aldrich, St. Louis, MO, USA) for 10–15 min at 37 °C and fixed in 4% paraformaldehyde (Cat^#^ P0099, Beyotime, Shanghai, China) overnight. Brain infarctions were then measured using 1.x ImageJ software [13]. The images were normalized to the contralateral hemisphere and expressed as a ratio according to the following formula: (area of the nonischemic hemisphere—area of the nonischemic tissue in the ischemic hemisphere)/area of nonischemic hemisphere [14].

### 2.4. Identification of the Microbiota Using Fecal 16S rRNA Sequencing

16S rRNA encodes a small subunit of the prokaryotic ribosome that is approximately 1542 bp in length. 16S rRNA is the most commonly used marker in bacterial phylotaxonomic studies. The main purpose of 16S rRNA gene sequencing is to study the species classification, species abundance and phylogeny in samples. In this study, fecal particles were collected on day 1, day 3 and day 7 after the first mild MCAO and two days after the second MCAO, and total microbial genomic DNA was extracted. The purity and concentration of the DNA were evaluated using 1.0% agarose gel electrophoresis and a NanoDrop^®^ ND-2000 spectrophotometer (Thermo Scientific Inc., Boston, MA, USA). Sequencing was conducted using an Illumina MiSeq PE300/NovaSeq PE250 platform (Illumina, San Diego, CA, USA).

### 2.5. Microbiota Sequencing Data Analysis Methods

A cluster analysis of operational taxonomic units (OTUs) was performed using RDP Classifier against the Silva v13816S rRNA gene database, and the confidence threshold was 70%. All bioinformatic analyses were completed using the Majorbio Cloud platform. The sample similarity of the microbiota was identified via a principal coordinate analysis (PCoA) based on Bray—Curtis dissimilarity, and the distinct difference in abundance (phylum to genera) of bacteria among the groups was determined via the linear discriminant analysis (LDA) effect size (LEfSe).

The default random forest algorithm was applied to analyze the importance of the microbiota [15]. Random forest is a classical and efficient machine learning algorithm based on a decision tree, which is a nonlinear classifier that can dig deep into the complex nonlinear interdependencies between variables, especially for microbial community data that often present discrete and discontinuous distributions [16]. The analysis software used in this study is QIIME 2 (2019.4). The detailed analysis steps were as follows: by default, the unflattened ASV/OTU table was used, or the absolute abundance table of taxonomic units in phylum, class, order, family and genus generated by the resulting taxonomic level was used, and the “classify_samples_ncv” function in q2-sample-classifier was used to perform a random forest analysis and nested hierarchical cross-check. When the maximum number of samples in the group was not less than 12, 10-fold cross-validations were performed. When the maximum number of samples in the group was less than 12 and greater than or equal to 7, a 5-fold cross-test was performed. If the maximum number of samples in the group was less than 7, the cross-test multiple was set to this value minus 2.

### 2.6. Metabolic Pathway Differences Analysis

PICRUSt2 (Phylogenetic Investigation of Communities by Reconstruction of Unobserved States) is a software that predicts the functional abundance of a sample based on the abundance of marker gene sequences in the sample (https://github.com/picrust/picrust2/wiki, (accessed on 14 September 2022)). (1) First, the 16S rRNA gene sequences of known microbial genomes were aligned to construct an evolutionary tree, and the gene function profiles of their common ancestors were inferred. This step was performed by the software. (2) The 16S rRNA feature sequence was aligned with the reference sequence to construct a new evolutionary tree. (3) Using the Castor hidden state prediction algorithm, the nearest sequence species of the characteristic sequence was estimated according to the copy number of the gene family corresponding to the reference sequence in the phylogenetic tree, and then the copy number of the gene family was obtained. Note that, when calculating the nearest sequence species index (NTSI) for each sequence, if the sequence was NTSI > 2 by default, it was excluded in subsequent analyses. (4) Combined with the abundance of the characteristic sequences of each sample, the copy number of the gene family of each sample was calculated. Note that, here, we used hierarchical processing; that is, for the gene family of each feature sequence, we added the species information of the sequence and output the results in layers to achieve the corresponding analysis of function and species. (5) Finally, the gene family was “mapped” to various databases, and the existence of metabolic pathways was inferred by default using MinPath to obtain the abundance data of the metabolic pathways in each sample. Once the data on the abundance of metabolic pathways were obtained, metabolic pathways that were significantly different between groups were identified using the metagenomeSeq method. Using the normalized pathway/group abundance table, according to the grouping situation and the tutorial example of metagenomeSeq, the fitFeatureModel function was applied to use the zero-inflated log-normal model to fit the distribution of each pathway/group, and the fitting results of the model were used to determine the significance of the difference.

### 2.7. Statistical Analysis

Significant differences among the groups were determined using two-tailed unpaired and paired Student’s *t* tests. For experiments with one treatment and more than two groups, a one-way analysis of variance (ANOVA) was applied using GraphPad Prism 7.0 (GraphPad Software, Boston, MA, USA).

## 3. Results

### 3.1. Recurrent MCAO Increased Brain Infarction and Decreased Mouse Survival

In this study, mice were randomly divided into three groups: a group subjected to sham surgery, a group subjected to single mild MCAO or a group subjected to mild stroke followed by another MCAO 6 days later (Figure 1A). Compared to the mice that received one MCAO surgery, those that received two MCAO surgeries had a much worse survival rate. Specifically, while 75% of the mice survived after one-time surgery, only 25% of the mice survived after two MCAO procedures (Figure 1B). According to the TTC staining, the mice with one mild MCAO had 20% infarction, while the infarction size in the mice with two MCAO events increased up to 40%~50% (Figure 1C,D).

### 3.2. Ischemic Stroke Reshaped the Composition of the Gut Microbiota in a Mouse Model

To investigate whether brain infarction eventually affects the gut microbiota, we collected fecal samples on day 1, day 3 and day 7 after the first MCAO and 2 days after the second MCAO (day 9), as well as from the sham control (Figure 1A). Based on the 16S rRNA sequencing data, we observed that, after secondary MCAO, there was a dramatic decrease in the amount of gut microbes at the family, genus and species levels (Figure 2A and Table 1). At the phylum level, several major types of bacteria, such as *Firmicutes*, *Bacteroidetes*, *Proteobacteria*, *Actinobacteria*, *Deferribacteraceae*, *Verrucomicrobia* and *Tenericutes,* were also commonly detected in other studies [17] (Figure 2B). The phyla *Firmicutes* and *Bacteroidetes* are the dominant microbiota in mice [18,19]. Notably, in the sham group, the ratio between *Firmicutes* and *Bacteroidetes* was nearly 1:1. After single mild MCAO, this ratio did not significantly change; however, in the recurrent MCAO mouse model, the ratio increased to 87.83%, and *Bacteroidetes* accounted for only 8.46% of the total microbiota, suggesting that, after recurring MCAO, the balance of the gut microbiota was disrupted (Figure 2B). In addition, compared to the sham control, the most abundant genus was *Lactobacillus*, but there were fewer *Firmicutes* in the recurrent MCAO mice. Furthermore, there was an increased abundance of *Akkermansia muciniphila* on day 7 after single MCAO (Figure 2C).

Further analysis led us to observe whether the species richness and diversity of the gut microbiota significantly changed after ischemic stroke. We applied four independent analysis methods to reveal the diversity of the microbiota, namely, Chao1, Shannon, Simpson and Observed_species. It is noteworthy that the diversity of the gut microbiota in the recurrent MCAO mice was significantly downregulated compared to that of the sham control and the mice subjected to one-time MCAO (Figure 3A). In addition, as evaluated by the PCoA analysis, the recurrent MCAO group was clearly separated from the other four groups, indicating significant changes (Figure 3B and Table 2). Next, we applied the LEfSe analysis of 16S rRNA sequencing to reveal the gut microbiota composition. The data showed that *Firmicutes* had an absolute advantage in the stroke group mice. Specifically, *Lactobacillus*, *Lactococcus*, *Streptococcus*, *Terribacillus* and *Bacillus* were more abundant in the stroke group. Similarly, the relative abundance of *Candidatus Saccharimonas* in the phylum *Proteobacteria* was greater in the sham control group (Figure 4A).

Having explored the differences in the composition of the microbes, we also needed to determine which species are mainly responsible for these differences. Thus, the random forest machine learning method was applied. As shown in Figure 4B, *Lactobacillus* was ranked the first as the most important microbiota after ischemic stroke [8]. *Lactococcus*, *Parabacteroides* and *Ruminiclostridium* were also listed as the top microbiota (Figure 4B).

### 3.3. Ischemic-Stroke-Induced Changes in the Gut Microbiota Reshaped Metabolic Characteristics

The gut microbiota is commonly accepted as an essential regulator of metabolism, considering that various microorganisms are involved in different biological processes, as well as the elucidation of metabolic pathways. Therefore, based on the 16S sequencing data, we further explored how the altered gut microbiota influences metabolic pathways (Figure 5 and Figure 6). Our data showed that, compared to the mice in the sham control group, for the mice in the one-time MCAO group, the most dramatic changes occurred 3 days after MCAO. There were more than 40 metabolic pathway changes after 3 days, and fewer than 20 metabolic pathways were altered on day 1 and day 7 after mild stroke compared with the sham control group (Figure 5A). Among these pathways, five metabolic pathways were significantly upregulated on all three days after MCAO compared to the sham control group (Figure 5B). These metabolic pathways included L-arginine degradation, polymyxin resistance, 3-hydroxyphenylacetate degradation, the superpathway of L-threonine metabolism and glycogen degradation (Figure 5B).

When the mice underwent secondary MCAO, the changes in gut microbiota-triggered metabolic shifts were much more dramatic. Approximately 50 metabolic pathways changed when compared to single mild MCAO and the sham control (Figure 6A). Notably, 14 metabolic pathways were found to be commonly downregulated when comparing the recurrent MCAO group to the other four groups (Figure 6B). In addition, 29 metabolic pathways were found to be altered when comparing recurrent MCAO to day 1 after single mild MCAO or day 7 after single MCAO (Appendix A), indicating that recurrent MCAO had a strong influence on the gut microbiota and further reshaped gut metabolism.

## 4. Discussion

There is increasing evidence highlighting the significance of the gut microbiota in reshaping microenvironmental metabolism and being involved in pathological processes in various neurological disorders, including ischemic stroke [7,8,20,21]. In the present study, we established a recurrent MCAO mouse model to determine the details of the changes in the gut microbiota on day 1, day 3 and day 7 after single mild MCAO, as well as after secondary MCAO. This investigation presented a systematic picture of microbiota dysbiosis after single and recurring ischemic stroke and showed potential changes in metabolism in mice.

Microbiota dysbiosis has been shown in both animal models and clinical samples after ischemic stroke [19,22]. Based on previous findings, the *Firmicutes*/*Bacteroidetes* ratio has been commonly accepted as a hallmark of aging and is significantly associated with ischemic stroke in mouse models [10]. From a clinical perspective, the assessment of the *Firmicutes*/*Bacteroidetes* ratio in the human gut microbiome has also been considered an essential marker. A higher abundance of *Firmicutes* and a lower abundance of *Bacteroidetes* are often manifestations of serious diseases [23]. In this study, we showed that the *Firmicutes*/*Bacteroidetes* ratio experienced a dramatic decrease after recurrent MCAO, along with enlarged infarction, indicating significant microbiota dysbiosis (Figure 2). These data provide experimental evidence demonstrating that the *Firmicutes*/*Bacteroidetes* ratio fits not only the single ischemic stroke model but also the recurrent stroke mouse model.

We also noted a significant increase in *Lactobacillus* and *Lactococcus* after recurrent MCAO compared with the sham control group (Figure 4B). Clinically, *Lactobacillus* was shown to be dramatically upregulated in 41 stroke patients versus 40 healthy controls in a Japanese cohort [24]. Li et al. also reported similar results based on a Chinese cohort including 79 stroke patients and 98 healthy controls [25]. Both *Lactobacillus* and *Lactococcus* belong to the lactic acid bacteria (LAB) group [26]. Lactate is mainly fermented to butyrate by butyrate-producing bacteria (BPB). Thus, when the butyrate level decreases, which is beneficial for lowering the inflammatory response and protecting the brain, LAB abundance compensates to produce more lactate for fermentation to butyrate [25].

The α-diversity is a metric for microbiome diversity within a sample. In our current study, although, compared to the sham group, we did not observe dramatic changes in diversity after single mild MCAO, there were significant changes after secondary MCAO. These results are consistent with published data based on clinical samples. For instance, there were five independent clinical studies, in which the cohort size was approximately 150–200 people, including both stroke patients and controls, revealing that, after ischemic stroke, the index of α-diversity, such as the Chao1, Simpson or Shannon indices, did not change [25,27,28,29]. It is notable that other clinical studies have reported contradictory results. In a larger scale cohort study, Yin et al. showed that Chao1 and observed species were increased based on 322 stroke patients and 231 controls in a Chinese cohort [30]. In cohort from the Netherlands, Haak et al. reported decreased Shannon and Simpson indices in 349 stroke patients and 51 controls [31]. The possible reasons for this difference may be due to the severity of the ischemic stroke and the sample size. When the ischemia is mild and the population size is relatively small, it may be hard to notice changes. Nevertheless, when the ischemia is severe, such as the recurrent MCAO in our current study, the α-diversity experiences a sharp decrease.

In the present study, we further explored the potential changes in the gut microbiota involved in metabolic pathways. Compared to various time points after single MCAO, in the sham group, one of the top pathways increased in the L-arginine degradation pathway. The amino acid L-arginine serves as a substrate for nitric oxide synthase. The derivatives of arginine, such as asymmetric and symmetric dimethylarginines, are regarded as markers of endothelial dysfunction and have been implicated in vascular disorders [32,33]. Clinically, in a Danish patient cohort, L-arginine was significantly higher in patients with acute ischemic stroke (*n* = 55) than in healthy individuals (*n* = 45) [34]. 3-Hydroxyphenylacetic acid degradation is another metabolic pathway affected by ischemic stroke. 3-Hydroxyphenylacetic acid (3-HPAA) is formed by the gut microbiota. Dias et al. showed that 3-HPAA relaxed precontracted porcine coronary artery segments via a mechanism partially dependent on endothelium integrity and further controlled the blood-pressure-reducing flavonoid metabolite [35]. Combining these two metabolic pathways could indicate that the alteration of the gut microbiota after acute ischemic stroke could regulate metabolites that further influence endothelial cells and blood pressure [36].

Furthermore, our data showed that more dramatic changes in metabolic pathways occurred after secondary MCAO. Notably, two pathways, L-tryptophan degradation and catechol degradation, were enriched twice in the common metabolic pathway (Figure 6B). It is known that the stroke-induced inflammatory response could upregulate the kynurenine (KYN) pathway for tryptophan (TRP) oxidation [37]. In an experimental stroke model, the L-kynurenine/aryl hydrocarbon receptor pathway was shown to mediate brain damage after stroke [38]. Furthermore, in a Belgian cohort, the plasma concentrations of tryptophan were measured in 149 stroke patients at admission, at 24 h, at 72 h and on day 7 after stroke onset. Their investigation showed that the activity of the kynurenine pathway for tryptophan degradation in acute ischemic stroke correlates with stroke severity and long-term stroke outcome [37]. Our results show that, after secondary stroke, L-tryptophan degradation was further elevated compared with single ischemia. The other enriched metabolic pathway was catechol degradation (Figure 6B). Although there are very few studies showing a direct relationship between catechol and ischemic stroke, many clinical studies have shown that catecholamines are associated with an increased risk of poststroke infections [39,40] due to catecholamine-induced immune cell apoptosis [41]. Hall et al. showed that genetic variation in catechol-O-methyltransferase (COMT), a key enzyme in estrogen and catecholamine metabolism, and the COMT rs4818G allele was associated with lower cardiovascular disease risk and lower fibrinogen levels [42]. Therefore, the decreased degradation of cardiovascular disease in recurrent MCAO may suggest that higher levels of catecholamines induce immune cell death, which may lead to worse outcomes.

Notably, there are still some limitations to the current study. For instance, using the same collected fecal sample to perform metabolic screening may provide more solid correlations between fecal microbiota and metabolism. In addition, Wang et al. noted that, compared to male mice, the gut microbiota from female mice could decrease proinflammatory cytokines after ischemic stroke [10]. Modulating the gut microbiota through methods such as fecal microbiota transplantation (FMT) has shown its therapeutic potential. Compared to male mice, female mice had less infarction and better behavior test performance. Male mice that received female microbiota through FMT received the protective characteristics of females [10]. Therefore, more research should be conducted using female and aged mice, and it should be examined whether transplanting fecal samples from healthy mice transmits the positive modulated factor from the gut microbiota. Nevertheless, our current study provides the first systemic picture of fecal microbiota changes. Based on a bioinformatics analysis, we also reviewed the microbiota-related metabolic shifts from the sham control group to the single MCAO and secondary MCAO groups in a mouse model.

## 5. Conclusions

In the present study, we systemically characterized the gut microbiota profile after single mild ischemic stroke and recurrent stroke using an experimental mouse model. Our data showed that acute ischemic stroke, especially after secondary ischemia, decreased the biodiversity of the gut microbiota.

## Figures and Tables

**Figure 1 biomedicines-12-00195-f001:**
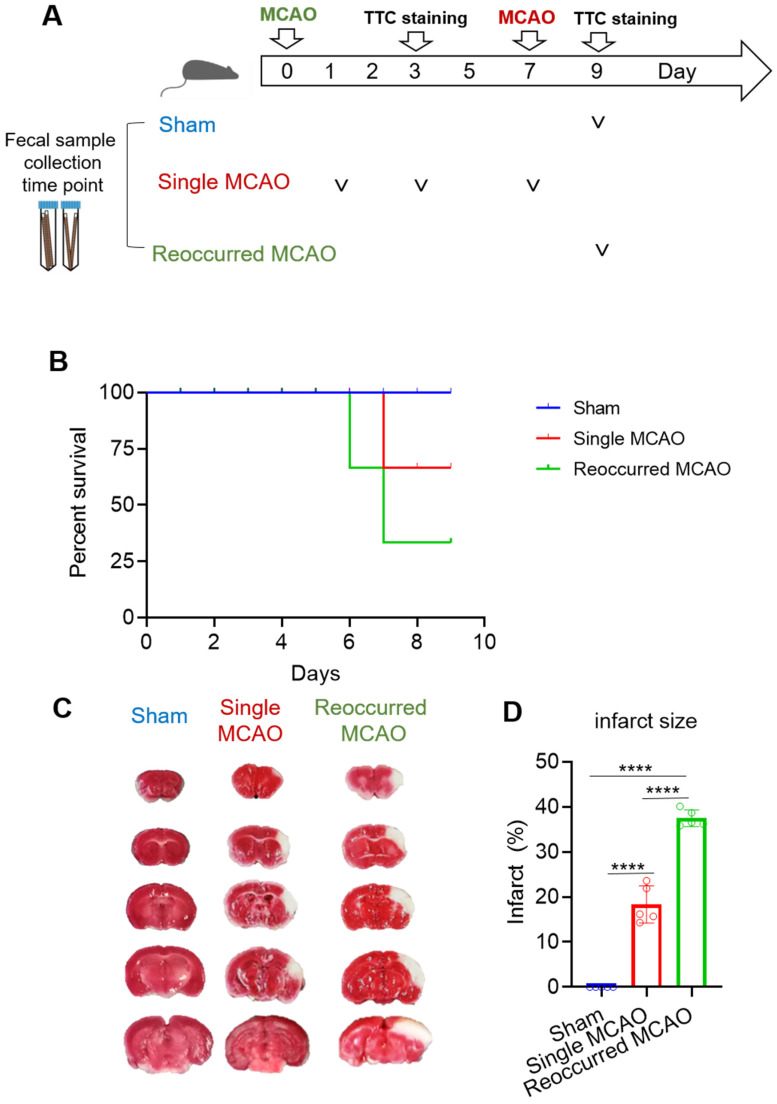
Recurrent ischemic stroke decreases the survival rate and enlarges the infarct size. (**A**) Schematic diagram of the experimental setup. Mice were randomly divided into three groups. To ensure the success of MCAO surgery, mouse brains were collected for TTC staining on day 3 after the first MCAO surgery (day 3) and 2 days after the second surgery (day 9). For the sham group, fecal samples were collected on day 9. For the single MCAO group, fecal samples were collected on day 1, day 3 and day 7. For recurrent MCAO, fecal samples were collected on day 9. (**B**) Survival rate. Sham group *n* = 5; single MCAO group, *n* = 10; recurrent MCAO group, *n* = 8. (**C**) Representative images of the infarction area identified via TTC staining (left) and quantification of brain infarction (right), *n* = 5/group. (**D**) Quantification of the infarct size. Blue represents the sham group, red represents the single MCAO group, and green represents the recurrent MCAO group. One-way ANOVA was performed, **** *p* < 0.0001.

**Figure 2 biomedicines-12-00195-f002:**
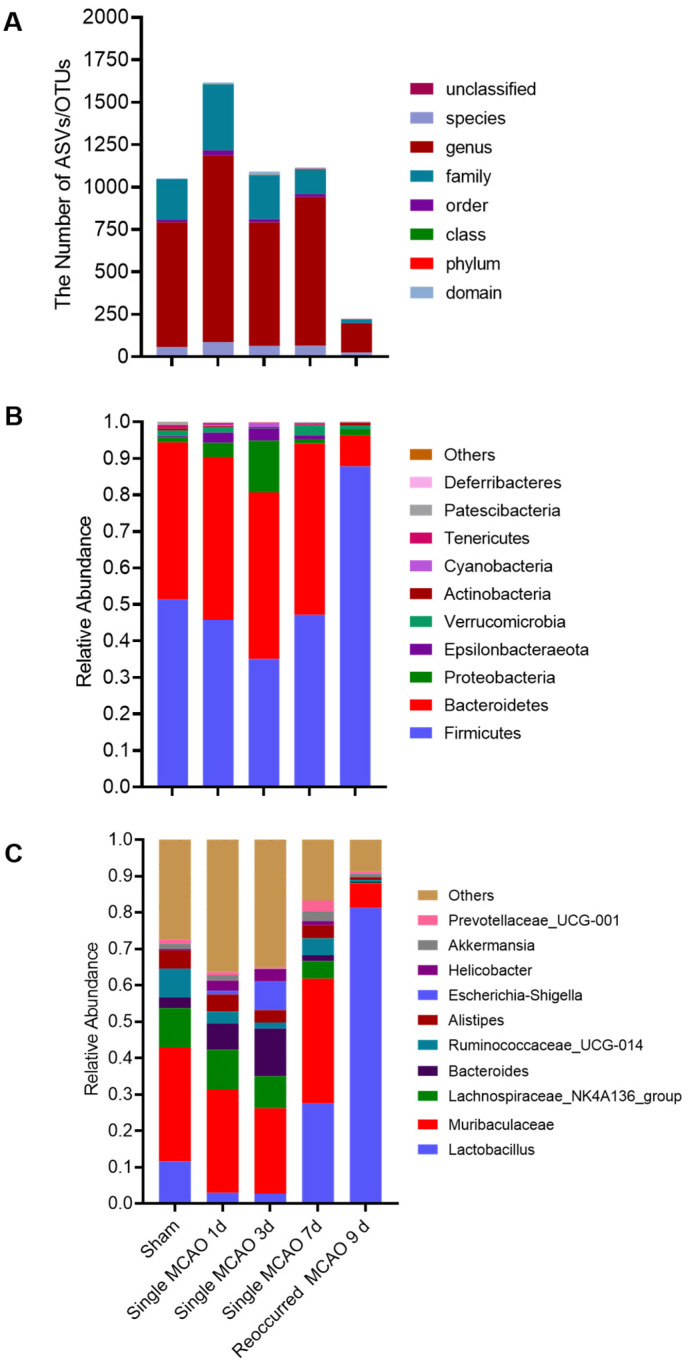
The composition of the gut microbiota population in the sham group, single MCAO surgery group and recurrent MCAO group. (**A**) The number of microbes. Taxa abundance of each group at various taxonomic levels. (**B**) Ischemic stroke induces changes in gut microbiota components at the phylum level and (**C**) genus level.

**Figure 3 biomedicines-12-00195-f003:**
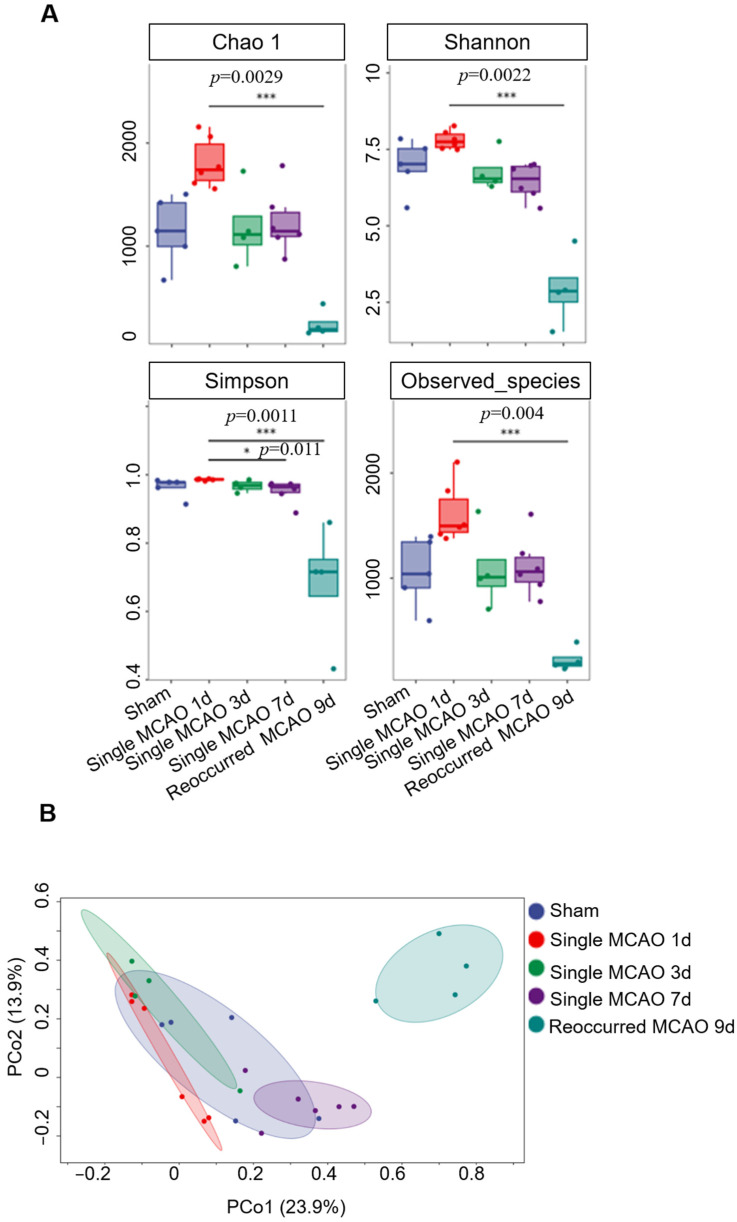
Recurrent MCAO significantly altered the alpha diversity of the gut microbiota. (**A**) Four alpha diversity index analyses, namely, Chao 1, Shannon, Simpson and observed species. The blue box represents the sham group, the red box represents fecal samples collected from mice 1 day after single MCAO, the green box represents fecal samples collected from mice 3 days after single MCAO, the purple box represents fecal samples collected from mice 7 days after single MCAO, and the cyan box represents fecal samples collected from mice 2 days after recurrent MCAO. (**B**) PCoA analysis of the microbiota among each group.

**Figure 4 biomedicines-12-00195-f004:**
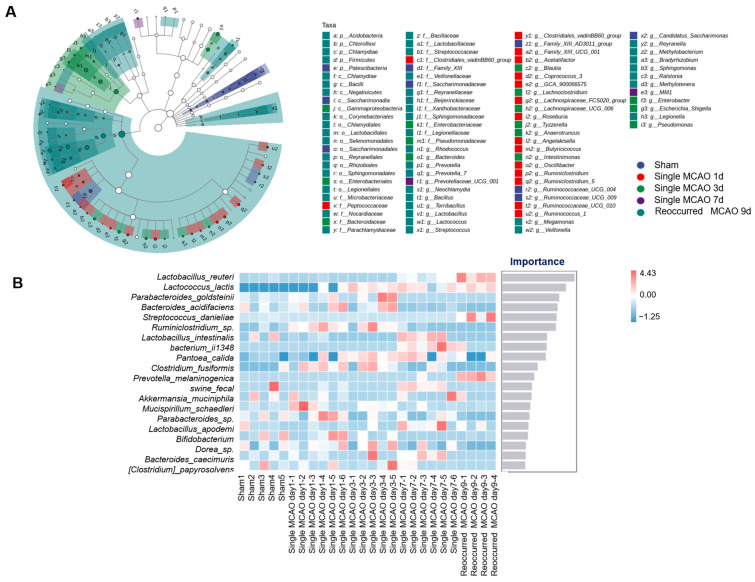
Differentially abundant gut microbiota among the groups. (**A**) LEfSe analysis shows the classification of the microbiota among the sham control group, single MCAO surgery group and recurrent MCAO group. The blue dots represent the sham group, the red dots represent fecal samples collected from mice 1 day after single MCAO, the green dots represent fecal samples collected from mice 3 days after single MCAO, the purple dots represent fecal samples collected from mice 7 days after single MCAO, and the cyan dots represent fecal samples collected from mice 2 days after recurrent MCAO. (**B**) In the diagram, the abscissa of the histogram is the importance score of the species to the classifier model, and the ordinate is the ASV/OTU (default) or the taxon name at the phylum, class, order, family, genus or species level. The heatmap shows the distribution of the abundance of these species in each sample/group. From top to bottom, species have diminished importance to the model, and these highest importance species can be considered to be markers of differences between groups.

**Figure 5 biomedicines-12-00195-f005:**
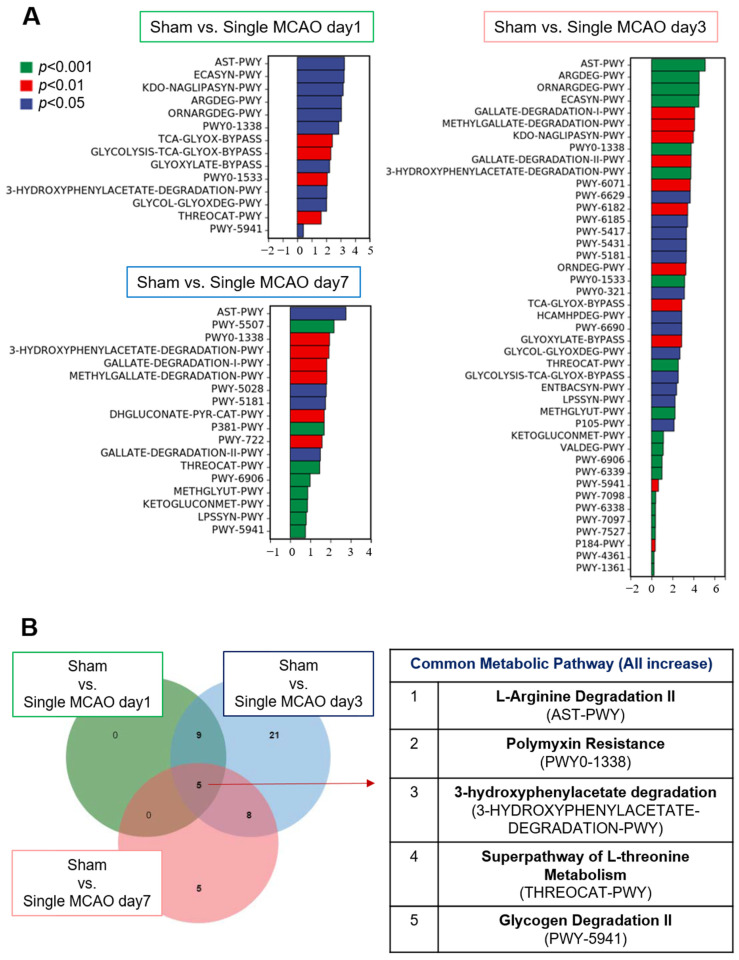
Metabolic pathway difference analysis between sham control mice and ischemic stroke mice. (**A**) Metabolic pathway differences analysis. The upper left shows the difference between sham control mice versus mice after single MCAO for 1 day; the lower left shows the difference between sham control mice versus mice after single MCAO for 3 days; the right shows the difference between sham control mice versus mice after recurrent MCAO for 2 days. (**B**) Venn diagrams show the commonly altered metabolic pathway numbers when ischemic stroke occurred (**left**) and the detailed pathway list (**right**).

**Figure 6 biomedicines-12-00195-f006:**
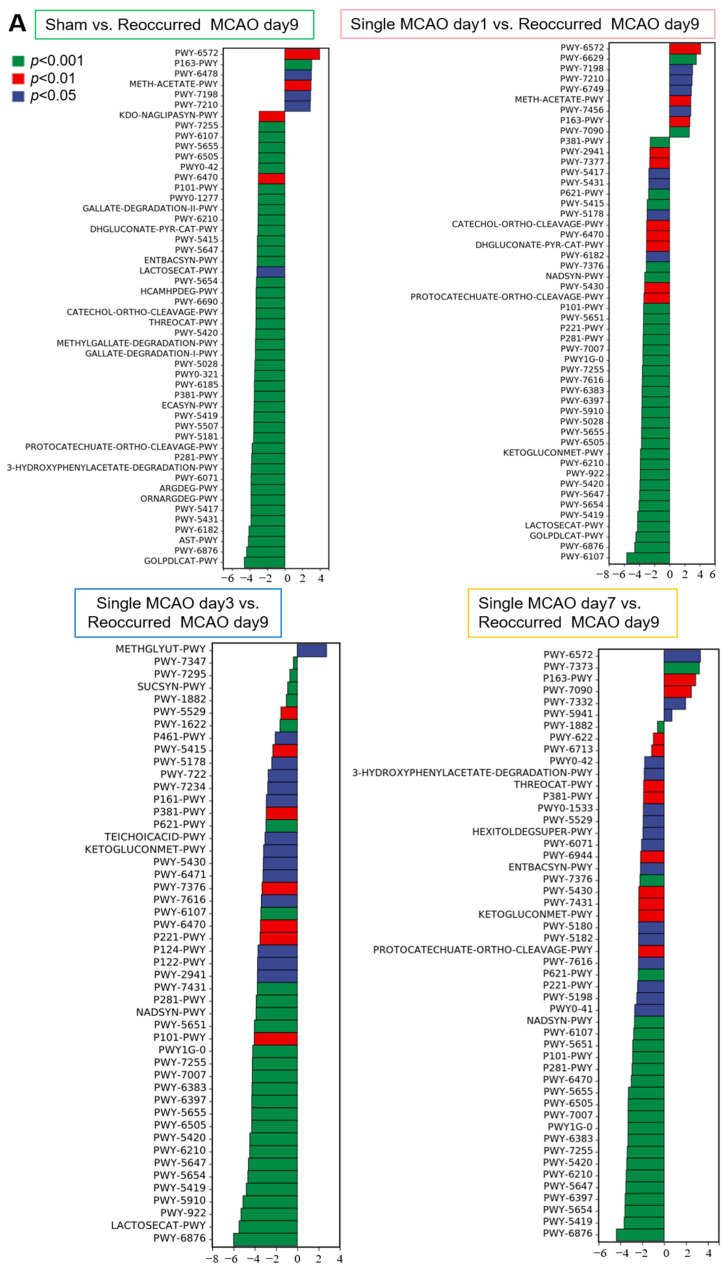
Metabolic pathway differences analysis among the different groups. (**A**) Metabolic pathway differences analysis. The differences in metabolic pathways enrichment between the recurring MCAO group and the sham control group were as follows: mice after single MCAO for 1 day, mice after single MCAO for 3 days and mice after single MCAO for 7 days. (**B**) Venn diagrams showing the commonly altered metabolic pathway numbers when ischemic stroke occurred (**upper**) and the detailed pathway list (**lower**).

**Table 1 biomedicines-12-00195-t001:** Statistics of taxonomy annotation results.

Group	Domain	Phylum	Class	Order	Family	Genus	Species
Sham	5.8	1.6	0.4	14.4	240.8	733.4	59.2
Single MCAO 1d	11	2.83	0.83	30.17	387.17	1099.83	87.17
Single MCAO 3d	18.25	1.25	2	19.25	260.75	728.25	63
Single MCAO 7d	7.17	6.17	0.67	17.17	142.67	878.83	64
Recurrent MCAO 9d	0.25	1.5	1	2.75	22.75	172	25.25

**Table 2 biomedicines-12-00195-t002:** Table of differences between groups.

Group 1	Group 2	*p* Value
Sham group	Single MCAO 1d	0.101
Sham group	Single MCAO 3d	0.022
Sham group	Single MCAO 7d	0.022
Sham group	Recurrent MCAO 9d	0.01
Single MCAO 1d	Single MCAO 3d	0.098
Single MCAO 1d	Single MCAO 7d	0.005
Single MCAO 1d	Recurrent MCAO 9d	0.002
Single MCAO 3d	Single MCAO 7d	0.004
Single MCAO 3d	Recurrent MCAO 9d	0.006
Single MCAO 7d	Recurrent MCAO 9d	0.007

## Data Availability

The data presented in this study are available on request from the corresponding author.

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
