# Peer review of "Systemic Characterization of the Gut Microbiota Profile after Single Mild Ischemic Stroke and Recurrent Stroke in Mice"

_biomedicines, 2024, doi:10.3390/biomedicines12010195_

Round 1

Reviewer 1 Report

Comments and Suggestions for Authors

Dear Sirs,

this is a very interesting original research regarding the alterations in the gut microbiota regarding stroke and recurrent stroke in an animal model. The study design is well presented; the references are up to date and the illustrations are very nice. However, the authors should include a Conclusion section at the end of the paper, stating their main conclusions. In addition, in the discussion section, more work needs to be prepared. For example, the paper would be improved by adding more comments regarding the modulations in the gut microbiota synthesis. The F/B ratio modification is indeed a very interesting finding, but further discussion on the synthesis of the gut microbiota in animals with recurrent stroke  per se should be added. Furthermore, in the abstract, the conclusion should be more specific and analytical. After these modifications, the paper would be accepted for publication, in my opinion.

Comments on the Quality of English Language

English language needs minor editing.

Author Response

Author’s response: We thank reviewer for the positive comments and suggestions.

Question 1. The authors should include a conclusion section at the end of the paper

Answer: Thank you, the conclusion section has been added into the end of this manuscript (highlight in yellow).

Question 2. In the discussion section, more work needs to be prepared. For example, the paper would be improved by adding more comments regarding the modulations in the gut microbiota synthesis. The F/B ratio modification is indeed a very interesting finding, but further discussion on the synthesis of the gut microbiota in animals with recurrent stroke per se should be added.

Answer: Modulation of gut microbiota is an important aspect, usually there are two ways: transfer of entire microbial communities (by using FMT) or transfer of a single microbial taxon. This method are needs to be more precise with known implement true “precision” modulation of the gut microbiota. We have added this part in discussion section as following: (and highlight in yellow in the manuscript)

“Modulating the gut microbiota through methods such as fecal microbiota transplanta-tion (FMT) has shown its therapeutic potentials. Compared to male mice, female mice had less infarction and better behavior test performance. Mice received female micro-biota through FMT received the protective characteristics of females to males [10].Therefore, more research should be done using female and aged mice, and checked whether transplant the fecal sample from healthy mice transmitted the positive mod-ulated factor from gut microbiota.”

Question 3. , in the abstract, the conclusion should be more specific and analytical.

Answer: Thank you, the abstract has been modified as suggested (highlight in yellow) as following:

“In the present study, using an experimental mouse model, we showed that acute ischemic stroke, especially reoccurred ischemia, significantly decreased the diversity of the gut microbiota.”

Reviewer 2 Report

Comments and Suggestions for Authors

Thank you for the interesting report using mice to search on the relation ship betwewn ischemic stroke and microbiota.

Abstract is very easy to understand and Introduction is also good to read. 

Doyou have any evidence about human between microbiota and stroke outcome? Please mention not only mouse but also human research.

What is the clinilcal application of this finding? Stool transplantation? Biophylaxis?

Do you have any findings between sarcopenia, stroke, and microbiota? PMID: 35277046 

Author Response

Authors’ response: We would like to thank the reviewer for the detailed comments. We have revised our manuscript accordingly. Thank you very much for these suggestions, which has significantly improved the quality of this manuscript.

Question 1. Please mention not only mouse but also human research. What is the clinilcal application of this finding? Stool transplantation? Biophylaxis?

Answer: We agree with reviewer that it is important to adding patient data in the manuscript. In the discussion section (paragraph 2-3 on page 16) we have discussed the patient clinical data, and we agree with reviewer, that mention and discussion human research is an essential factor for evaluate the value of the study.     Furthermore, as reviewer suggested, we also added stool transplantation (FMT) into the discussion section (highlight in yellow) as following:

“Modulating the gut microbiota through methods such as fecal microbiota trans-plantation (FMT) has shown its therapeutic potentials. Compared to male mice, female mice had less infarction and better behavior test performance. Mice received female microbiota through FMT received the protective characteristics of females to males [10].Therefore, more research should be done using female and aged mice, and checked whether transplant the fecal sample from healthy mice transmitted the positive mod-ulated factor from gut microbiota.”

Question 2. Do you have any findings between sarcopenia, stroke, and microbiota? PMID: 35277046 

Answer: Thank you for this suggestion. The paper “Temporal Muscle and Stroke—A Narrative Review on Current Meaning and Clinical Applications of Temporal Muscle Thickness, Area, and Volume” published in Nutrients (PMID: 35277046) is very interesting. We haven’t looked at the relationship between temporal muscle and stroke in the current study, but it would be a very interesting areas to study, and thanks again for your suggestion.

Reviewer 3 Report

Comments and Suggestions for Authors

The authors collected fecal samples at various time points to provide a systematic picture of how the microbiota changes after the 1st and 2nd strokes and the potential influenced on metabolic pathways. The discussion integrates the results with existing literature, and potential mechanisms are proposed. This manuscript is well written and can be published after a few corrections of editing errors and minor issues.

Below are my comments:

In clinical practice, recurrent stroke often occurs years after the first insult. What is the clinical implication of 2 strokes in rats occur 6 days apart?

The authors found significant differences between phylum and genus. What were the statistical figures in Fig 2B and 2C (line 214-222)?

Acronyms should be used correctly and should not be used for medical term with only one appearance. For example, ADMA and SDMA were noted in line 362.

Some typo and editing errors: line 367, be Dias et al.; line 389, be Hall et al.;

Line 333-334: “We also noticed a significant increase in Lactobacillus and Lactococcus (Figure 2 and 333 Figure 4).” Unclear syntax here was noted.

Reference style should be consistent. For example, the style of ref 1 is different from ref 3.

Comments on the Quality of English Language

English is fine in this manuscript.

Author Response

Authors’ response: We sincerely thank the reviewer for the positive comments and suggestions, which are really helpful for us to improve the quality of this manuscript.

Question 1. In clinical practice, recurrent stroke often occurs years after the first insult. What is the clinical implication of 2 strokes in rats occur 6 days apart?

Answer: Thank you for asking this issue. There is increasing evidence showing that brain injury is associated with increased risk of stroke recurrence (Hill et al., 2004 ,PMID: 15184607; Coutts et al., 2005,PMID: 15929051). There are published literatures indicated that a second stroke occur within 90 days in 10–20% of patients who have had an initial TIA, and in half of these patients the recurrent stroke occurs in the first week after the TIA (Johnston et al., 2004, PMID: 15051699; Correia et al., 2006,PMID: 16322498; Nguyen-Huynhet al. , 2007, PMID: 17522720). In addition, based on our pervious study, the immune system in MCAO mice, could recovery after 7 days after 1st surgery (PMID: 32908941). Considering the above points, we chose this time point in current study.  

Question 2. The authors found significant differences between phylum and genus. What were the statistical figures in Fig 2B and 2C (line 214-222)?

Answer: Thank you for pointing this issue. The Figure 2B and 2C showed the composition of gut microbiota at various level were different. And the details were listed in Table 1. However, whether the difference are significant is requiring further data to support (Figure 3 and 4). The relevant statistics (p-value) were summarized in Table 2.

Question 3. Acronyms should be used correctly and should not be used for medical term with only one appearance. For example, ADMA and SDMA were noted in line 362.

Some typo and editing errors: line 367, be Dias et al.; line 389, be Hall et al.;

Line 333-334: “We also noticed a significant increase in Lactobacillus and Lactococcus (Figure 2 and 333 Figure 4).” Unclear syntax here was noted.

Reference style should be consistent. For example, the style of ref 1 is different from ref 3.

Answer: These errors have been modified in the manuscript and highlight in yellow. The reference has been double checked to ensure all papers are listed in the same style. Thank you!
